# DataMesh+: A Blockchain-Powered Peer-to-Peer Data Exchange Model for Self-Sovereign Data Marketplaces

**DOI:** 10.3390/s24061896

**Published:** 2024-03-15

**Authors:** Mpyana Mwamba Merlec, Hoh Peter In

**Affiliations:** 1Department of Computer Science and Engineering, Korea University, 145 Anam-ro, Seongbuk-gu, Seoul 02841, Republic of Korea; mlecjm@korea.ac.kr; 2DAO Solution, Inc., 169, Yeoksam-ro, Gangnam-gu, Seoul 06247, Republic of Korea

**Keywords:** blockchain, data marketplace, data mesh, peer-to-peer data trading, self-sovereign data marketplace (SSDM), smart contracts

## Abstract

In contemporary data-driven economies, data has become a valuable digital asset that is eligible for trading and monetization. Peer-to-peer (P2P) marketplaces play a crucial role in establishing direct connections between data providers and consumers. However, traditional data marketplaces exhibit inadequacies. Functioning as centralized platforms, they suffer from issues such as insufficient trust, transparency, fairness, accountability, and security. Moreover, users lack consent and ownership control over their data. To address these issues, we propose *DataMesh+,* an innovative blockchain-powered, decentralized P2P data exchange model for self-sovereign data marketplaces. This user-centric decentralized approach leverages blockchain-based smart contracts to enable fair, transparent, reliable, and secure data trading marketplaces, empowering users to retain full sovereignty and control over their data. In this article, we describe the design and implementation of our approach, which was developed to demonstrate its feasibility. We evaluated the model’s acceptability and reliability through experimental testing and validation. Furthermore, we assessed the security and performance in terms of smart contract deployment and transaction execution costs, as well as the blockchain and storage network performance.

## 1. Introduction

In the evolving landscape of data-driven economies of scale, data emerges as a valuable digital asset ripe for trading and monetization. According to [1,2], projections indicate a surge in the global data sphere to 181 ZB by 2025, as shown in Figure 1a, alongside an anticipated revenue boost for the global big data market to 655.53 billion dollars by 2029, as shown in Figure 1b. The convergence of mobile cloud computing and communications, the Internet of things (IoT), artificial intelligence (AI), big data analytics, and blockchain technologies has created unprecedented economic prospects for individuals and organizations to capitalize on their data [3,4,5,6,7,8,9,10,11,12,13,14,15,16,17,18]. However, the path to effective monetization of data is riddled with challenges, mainly stemming from the limitations of traditional online data marketplaces [4,5,6,7,8,9,10,11,12]. To address these challenges, the establishment of peer-to-peer (P2P) marketplaces is imperative, facilitating direct transactions between data providers (sellers) and consumers (buyers) over the Internet [7,8,9,10,11,12,13,14,15,16,17,18]. A P2P data marketplace is an internet-based marketplace, also referred to as an electronic marketplace (e-marketplace) platform where users can connect to directly exchange, sell or buy data with or without the involvement of intermediaries [12,13,14]. In the inherently trust-challenged realm of the Internet, trusted third parties (TTPs) play a vital role in fostering trust and resolving disputes among transacting parties [15,16,17]. Powered by a community comprising data providers and consumers, P2P marketplaces require a robust infrastructure to enable secure, fair, transparent, and reliable transactions, along with seamless payment processing [10,11,12,13,14,15,16,17,18,19,20,21,22]. 

However, traditional data marketplaces are inadequate. Operating as centralized platforms, they lack the necessary levels of trust, transparency, fairness, accountability, and security [3,4,5,6,7,8,9,10,11,12,13,14,15,16,17,18,19,20,21,22]. Despite their extensive adoption, these centralized data marketplaces are often face vulnerabilities such as single points of failure (SPF) and fail to offer adequate ownership and consent control over data use [16,17,18,19,20], thus contravening data protection regulations such as the General Data Protection Regulation (GDPR) [23]. Furthermore, their reliance on TTPs leads to increased transaction costs, inequitable revenue distribution, and increased complexity [11,17,22]. Consequently, users remain uncertain about how their data are collected and used, leading to mistrust and insufficiency in data transactions [18,19,20,21].

The emergence of *data mesh* architecture [24,25], *blockchain* [26,27,28], and *decentralized P2P storage* technologies [29] offers a paradigmatic solution to these pressing problems. *Data mesh* is a new conceptual data architecture framework that emphasizes decentralized ownership and control, treats data as a product, and focuses on domain-driven design [24,25]. This approach enhances value extraction from data by overcoming the limitations of traditional centralized data systems across various business domains within or across large and complex organizations [25,30]. Moreover, data mesh advances the establishment of a self-serving data infrastructure that enables domain teams to access and process data autonomously [30,31]. The core properties of blockchain–decentralization, immutability, and tamper resistance–herald a new era of secure, reliable, and transparent transactions [26,27]. By eliminating the need for central authorities, blockchain significantly reduces transaction costs while increasing efficiency [28,32]. The implementation of smart contracts on blockchain networks enables automated, trustless transactions characterized by transparency, auditability, and immutability [21,28,32]. This transition to a blockchain-based framework marks a stride toward decentralized P2P data marketplaces, offering a more equitable, secure, reliable, and transparent environment for data exchange. Despite advances in blockchain and smart contract-based P2P data trading models [12,13,14,15,16,17,18,19,21,22,33,34,35,36,37,38,39,40,41,42,43,44,45,46,47,48,49,50,51], existing systems have yet to fully address the inherent challenges of traditional marketplaces, particularly in terms of decentralization, transparency, fairness, security, trust, user control over data ownership, and consent control over data. How can blockchain, smart contracts, and decentralized storage technologies be leveraged to build secure, reliable, and user-centric decentralized data marketplaces? 

To address these challenges, this study made the following contributions:We introduced *DataMesh+,* an innovative blockchain-powered P2P data exchange model for decentralized self-sovereign data marketplaces (SSDMs). DataMesh+ advances the data mesh concept by integrating blockchain and decentralized storage technologies to enhance decentralization in data trading. It prioritizes user control by employing blockchain-based smart contracts to enable fair, transparent, reliable, and secure data trading marketplaces and empowers users to be sovereign and retain full control over their data. Smart contracts execute self-enforcing agreements between buyers and sellers, facilitating trustworthy transactions among globally disparate and anonymous parties without relying on centralized TTPs.We leveraged the Ethereum blockchain [27] to build a prototype that validates the practicality and effectiveness of our approach. To achieve pseudo-anonymity, users are identified through *externally owned accounts* (*EOAs*) provided by the Ethereum blockchain, which are secured with private and public cryptographic key pairs. Data ownership is determined by public-private key pairs, digital signatures, and account addresses. Digital signatures authenticate participant identities by cryptographically verifying transaction origins and holding them accountable by providing verifiable proof of their involvement in blockchain-recorded activities. Smart contracts track the participant activities and enforce the consequences of their actions. The *InterPlanetary File System* (IPFS) [29] provides resilient and highly available data storage and sharing capabilities in a secure, decentralized, and censorship-resistant manner, thus increasing the robustness of the proposed framework.We provided a comprehensive literature review addressing the design considerations, principles, and challenges associated with the development of a decentralized P2P data marketplace based on blockchain.We have outlined the operational workflow and architectural framework of the proposed model and its implementation. We assessed the acceptability and reliability of the proposed model through experimental testing and validation. Furthermore, we evaluated the security and performance in terms of smart contract deployment and transaction execution costs, the blockchain, and storage network performance.

The subsequent sections of this article are structured as follows: Section 2 provides an in-depth exploration of the literature review and preliminary concepts; Section 3 describes the design and architecture of the proposed model; Section 4 discusses the model implementation and evaluation; Section 5 addresses the limitations and remaining challenges; and Section 6 concludes the paper and offers insights into potential future research directions.

## 2. Literature Review and Preliminaries

This section provides an overview of existing data marketplaces and trends towards blockchain- and smart contract-based P2P data trading approaches. Subsequently, it presents preliminary research concepts including blockchain technology, smart contracts, IPFS, access control mechanisms, cryptographic primitives, and digital signatures.

### 2.1. Online Data Marketplaces

Recent years have seen extensive research into online data marketplaces [3,4,5,6,7,8,9,10,11,12,13,14,15,16,17,18,19]. These platforms vary across multiple dimensions, including target industry, business model, underlying technology, system architecture, type of data and services offered, trading mechanisms, use cases, and functionalities [5,6,7,8,9,10,11,12,13,14,15,16]. Foundational studies by [3,4,5,6,7,8,9,10,11,12] established a conceptual foundation and explored the concept of data as a commodity in a digital economy. In addition, refs. [5,10,11] present comprehensive definition and classification frameworks for data marketplaces that consider certain characteristics, including value propositions, market positioning, market access, and control, and data governance [16,17,18]. Furthermore, they address system architecture, integration, data acquisition, matching mechanisms, transformation, pricing, and revenue models [5,11,17]. The exploration of real-time data dynamics within these marketplaces and their impact on privacy and transaction efficiency is documented in [6,7]. The challenges and mechanisms of trading personal data, especially in IoT data marketplaces, are discussed in [8]. Hatamian [9] identified the technological barriers that hinder the development of IoT data marketplaces. Contributions from studies in [10,11] have enriched the understanding of data marketplace business models and introduced a taxonomy that explains the economic fundamentals of data trading. In [12], an architectural perspective for P2P data monetization in the context of fog computing is presented. 

However, traditional data marketplaces, characterized by their centralized architecture, fail to deliver the required level of trust, transparency, fairness, accountability, and security [5,6,7,8,9,10,11,12,13,14,15,16,17,18,19,21,22]. Moreover, users face challenges regarding consent and ownership control over their data to protect privacy [7,14,15,16,17,18,19,20], as mandated by data protection regulations, such as GDPR [23]. Reliance on TTPs in existing centralized marketplaces increases transaction costs and creates friction in revenue sharing [8,14,15,16,17,18,19,20,21,22]. Due to insufficient transparency, fairness, and user consent control in centralized marketplaces, users remain unaware of the collection, extent, timing, and recipients of their data sales [20,21,22]. Moreover, centralized architectures encounter SPF problems because administrators have unrestricted power and authority over user data. Consequently, these systems become prime targets for attackers seeking economic gains.

### 2.2. Data Mesh, Blockchain and Smart Contracts

Data mesh architecture [24,25], blockchain technology [26,27,28], and decentralized P2P storage networks [29] offer diverse features that P2P SSDMs can use to address the problems and limitations of existing centralized systems. Data mesh represents a novel approach to data architecture, transitioning from traditional centralized data management systems to a decentralized and domain-centric model [24,25]. This concept, pioneered by Zhamak Dehghani [24], revolves around treating data as a product and is based on four main pillars: (1) domain-oriented decentralized data ownership, (2) data as a product, (3) self-managed data infrastructure, and (4) federated computing management. The objective is to decentralize and control data ownership by distributing it among various business units within or between organizations. Each division regards its data as a product that promotes better alignment with business needs and enhances flexibility and scalability [30,31]. Although this approach enhances organizational agility and democratizes data access, it remains independent of the underlying technologies for data storage and transactions.

The concept of blockchain originally emerged as the technology supporting the Bitcoin system [26], a P2P electronic payment system that uses a public, shared, and immutable ledger to record a continuously expanding list of transactions. A blockchain represents a type of distributed ledger technology (DLT) for recording a series of time-stamped transactions that are sequentially linked in blocks and cryptographically secured [26,27,28]. Altering transactions in a block requires retroactive updating of all preceding blocks and obtaining consensus from most network participants [26,27]. Ethereum, an open-source, decentralized blockchain platform, is distinguished by its incorporation of featuring smart contracts [27,32], which are computer programs running on the blockchain that are capable of automatic execution upon meeting predefined conditions. Smart contracts facilitate the automation, execution, control, or documentation of events and actions in accordance with contract terms [20,27,32]. They are essential for developing decentralized applications (DApps) that operate on the blockchain. In contrast to traditional applications, blockchain offers DApps enhanced security and transparency in transactions [28,32]. The use of smart contracts can improve efficiency, trust, and security across various of applications [20,21,22,26,32,33], including cross-border data trading and electronic payments. Ethereum incorporates a universal, Turing-complete, virtual state machine known as the *Ethereum Virtual Machine* (EVM) [27], specifically for the execution of smart contract code. Within an Ethereum network, two primary types of nodes exist: miner nodes and regular nodes. Regular nodes facilitate transaction forwarding within the network, while miners verify and validate transactions by mining new blocks [27,28]. To ensure the consistency of the blockchain, a consensus protocol synchronizes the state of the ledger for each node in the network. Ethereum supports multiple consensus protocols including *Proof-of-Work* (PoW), *Proof-of-Stake* (PoS), and *Proof-of-Authority* (PoA) [26,27,28]. 

Blockchain’s key features include decentralization, transparency, tamper-resistance, and immutability. These attributes are facilitated by publicly accessible and verifiable distributed ledgers and storage that are consistent and secure in P2P networks [26,27,28]. Furthermore, blockchain technology offers transnational anonymity, immediacy, validity, traceability, and persistence [27,28]. In addition, business terms agreement can be embedded in smart contracts, which execute autonomously on the blockchain, free from censorship or third-party interference [22,27,32]. Thus, blockchain eliminates the need for a central authority to validate transactions, yielding significant computational and cost efficiencies [33,34,35,36,37,38,39,40]. Through smart contracts, novel trustless intermediation mechanisms for decentralized data marketplace services can emerge [41,42,43,44,45]. By eliminating outdated trust-building mechanisms in conventional e-marketplaces, marketplace intermediaries or TTPs are eliminated, thereby lowering barriers to entry and transaction costs [33,34,35,36,37,38,39,40,41,42,43,44,45,46,47,48,49,50,51]. The cryptographic security of blockchain and the decentralized nature of P2P storage reduce the risks associated with centralized data storage [20,21,22,27,28,29], such as data breaches and unauthorized access.

### 2.3. Blockchain-Enabled Data Marketplaces

Various initiatives are progressing toward blockchain- and smart contract-based P2P data trading approaches [13,14,15,16,17,18,19,20,21,22,33,34,35,36,37,38,39,40,41,42,43,44,45,46,47,48,49,50,51]. Table 1 outlines the comparison between the proposed approach and previous studies, considering the main features of the proposed model. Studies in [13,14,15,33,34,35,36,37,38] establish the foundational concepts for blockchain-based decentralized data trading platforms, facilitating direct transactions between buyers and sellers without the involvement of TTPs. Specifically, [13,14] advocate blockchain-based P2P marketplaces for data trading, emphasizing preservation of data ownership. The authors of [15] address trustless transactions in decentralized service marketplaces. This aligns with the trend toward decentralized marketplace systems, as shown in [16,17,18], which provides insights into the landscape of data marketplaces and their business models. The research endeavors of [17,18,19,21] shift the focus toward ensuring transaction integrity and the governance structures of blockchain-based data marketplaces. The role of smart contracts in establishing dynamic consent management systems is highlighted in [20], which emphasizes user control in digital data marketplaces. Wang et al. [22] demonstrated the application of blockchain in ensuring fair payment for cloud storage auditing, signifying the convergence of blockchain across various data marketplace applications. The studies of [33,34,35] explored the decentralized nature of blockchain-based marketplaces, whereas [36,37] proposed systems for the efficient trading of big data. In [34,38], the design considerations and implementation challenges of such trading platforms are explored, encompassing aspects such as fairness, efficiency, security, privacy, and regulatory compliance.

The decentralized marketplace model for digital content proposed by [39] further illustrates the application of blockchain in content distribution. Detailed descriptions of the essential requirements for a secure blockchain-based data trading ecosystem are provided in [40], while [41] explores the integration of machine learning with blockchain for data trading. In addition, [42] exemplifies a system developed for big data trading using blockchain, where smart contracts support data matching, price negotiation, and reward allocation. In contrast, [43] envisions a marketplace for self-managing machines with accurate and trustworthy data sources. The decentralized nature of blockchain as a basis for IoT data marketplaces is examined in [44,45,46]. The IDMob system [47] serves as a practical demonstration of a blockchain data marketplace, while [48] explores a trustless marketplace for IoT data. In addition, [49] investigated the security and efficiency of blockchain-based data trading systems for the Internet of vehicles (IoV). They introduced an iterative double data trading auction system for IoV, aiming to maximize social welfare using a consortium blockchain. In [50], the role of decentralized autonomous organizations (DAOs) in the production of citizen-generated data. Finally, the case study of the Wibson data marketplace by [51] provides insights into decentralized and privacy-preserving data trading.

Despite notable advances in research, achieving a comprehensive solution to the challenges of decentralization, transparency, fairness, security, trust, user consent, and data ownership in data marketplaces remains elusive. Our research aims to bridge these gaps by proposing DataMesh+, a robust model and architectural framework that complements the Data mesh model [24,25] and addresses these critical issues. This study offers a substantial contribution with a comprehensive and nuanced approach to designing data exchange for user-centric, secure, transparent, fair, and trustworthy decentralized P2P SSDMs empowered by blockchain technology and decentralized P2P storage networks.

### 2.4. Cryptographic Primitives and Digital Signature

Cryptography algorithms are indispensable for ensuring the authenticity, confidentiality, integrity, and non-repudiation of data. We have focused on asymmetric cryptography, specifically public key cryptography [52,53], as it plays a crucial role in modern information system security. Public key cryptography uses unique keys that are easy to compute but are based on mathematical functions whose inverses are difficult to compute. These cryptographic functions facilitate the creation of tamper-proof digital signatures and mathematically secured secrets. *Elliptic curve cryptography* (ECC) is a variant of asymmetric cryptography based on discrete logarithmic problems defined by adding and multiplying the points of an elliptic curve [53].


*Elliptic curve digital signature algorithm* (ECDSA) [53] is used for signing and verifying transactions. Its key pairs are associated with certain domain parameters consisting of an elliptic curve *E* represented over a finite field ($F_{p}$). *E* is characterized by a base point *G* ∈ *E*($F_{p}$) [53]. In practice, the parameters of a domain *D* are defined as (*q*, *FR*, *a*, *b*, *G*, *n*, *h*), which include *q*, the field size, where *q* is equal to *p*, an odd prime number, or 2*^m^*. *FR* denotes the field specification for elements of $F_{p}$. Parameters *a* and *b* refer to the field elements of the elliptic curve *E* over $F_{p}$, as defined in Equation (1) for the case where *p* > 3 and Equation (2) for the case where *p* > 2 [53].



(1)
$$E(F_{p}):y^{2}+xy+y^{3}=x^{3}+ax+b$$



*G* is a finite point of the curve, defined by *x_G_* and *y_G_* in $F_{p}$ as *G* = (*x_G_, y_G_*) with prime order in *E*($F_{p}$). *n* is the order of *G* such that *n* > 2160 and *n* > 4$\sqrt{q}$ and *h* is the cofactor, defined as *h* = #*E*($F_{q}$)/*n* [53]. A domain can have parameters that are either shared by multiple entities or unique to a particular user.



(2)
$$E(F_{p}):y^{2}=x^{3}+ax+b$$



*Secp256k1 curve*: Similar to Bitcoin, Ethereum uses secp256k1 [27,52], an elliptic curve primitive used for cryptographic encryption. Secp256k1 is defined over $F_{p}$, where *p* = 2^256^ – 2^32^ – 977 with 256-bit prime order, as expressed in Equation (3) [52,53].



(3)
$$E(F_{q}):y^{2}=x^{3}+7$$


*Private and public keys*: The private key is an arbitrary 256-bit integer *k*, multiplied by a predefined generator point *G* over the elliptic curve to produce another point from which the public key *K* is derived, as defined in Equation (4) [52,53]. The ECDSA key pair generation and validation algorithms are detailed in [53].


*K = kG*
(4)


*Keccak256/SHA-3* (*Secure hash algorithm 3*): The Keccak256 hash function, also known as the *SHA-3* cryptographic hash algorithm [27], computes the hash value of data stored in a blockchain. Equation (5) computes the Keccak256 hash of a given memory input and returns a 32-byte size hash value.


*Keccak256(bytes memory) returns(bytes32)*
(5)


*Ethereum address*: Ethereum addresses comprise 40 hexadecimal characters, unique identifiers derived from the corresponding ECDSA public key or the contract’s Keccak256 hash function (specifically its last 20 bytes), as follows [27,52]. Note that when the address is computed, the prefix (hex) 04 of the public key is omitted.


*A(k) = B
_96..25_
(KEC(ECDSAPUBKEY(k)))*
(6)


In the context of P2P data exchange, digital signatures play a pivotal role in ensuring the authenticity, integrity, and non-repudiation of data transactions between peers [29,52,53]. A digital signature is a cryptographic technique used to verify the authenticity and integrity of messages, documents, or transactions [52,53]. These signatures leverage mathematical algorithms to generate a unique digital fingerprint or signature for each piece of data. The signature is generated using the signatory’s private key and can only be verified using the corresponding public key. In the Ethereum adaptation of ECDSA, transaction messages are signed by computing the Keccak256 hashes of the transaction input data encoded with recursive length prefixes (RLP) [52,53,54]. The result is a signature *S*, which is defined as follows [52]: *S = F_sig_ (F_keccak_
_256_
(m), k),*(7)
where *m* represents the transaction serialized message, *k* denotes the EOA’s signing private key, and *F_sig_* and *F_keccak256_* correspond to the respective signing and Keccak256 hash functions, respectively. The signature creation algorithm is described in [52,53,54].

The verification is the reverse operation of a signature creation function for given signature values *r* and *s*, and the signer’s public key *K* is used to compute a value *X* via Equation (8) [27,52]. *X* represents an elliptic curve point that serves as a temporary public key used for signature generation.
*X ≡ (u*_1_*G + u*_2_*K)*(8)

The detailed specification of the signature verification algorithm can be found in [27,52]. It operates by using the message *M*, the signature’s public key *K*, and the digital signature *S* as inputs to compute point *X*. A valid signature is determined if *x*, the coordinate of the calculated point *X*, is equal to *r*. Digital signatures enhance the security and reliability of P2P data exchange by enabling secure authentication, preserving data integrity, and establishing non-repudiation mechanisms between participating peers.

### 2.5. InterPlanetary File System and Access Control

IPFS [29] is a P2P and distributed file storage and sharing system that leverages content-addressing to identify unique content files in a global namespace. This scheme enables off-chain storage of large files and embeds hyperlinks for each content within immutable transactions by time stamping and backing up the content without including the data in a blockchain [29,55]. Cryptographic hash algorithms paired with distributed multi-hash tables ensure data integrity. The essential properties of IPFS include decentralization, security, censorship resistance, high reliability and availability, and low network latency. However, due to content identifiers (CIDs) being globally accessible upon upload and pinning, IPFS inherently lacks confidentiality. Therefore, ensuring data confidentiality requires file encryption before the ciphertext is stored in the IPFS nodes instead of plain text. Decentralized access control and consent management empower users with data ownership and access control [20,55,56]. With the role-based access control (RBAC) scheme [57,58], user access rights and privileges can be assigned and managed based on their role in the system, such as a seller or buyer. Therefore, the integration of IPFS, blockchain, and smart contract-based decentralized consent and access control systems will fulfill the above-mentioned essential characteristics of the proposed P2P data marketplace model.

## 3. DataMesh+: Proposed Secure and Reliable Decentralized P2P Data Exchange Model

In this section, we discuss the intricate details of the proposed model, outline its operational workflow and architectural framework, and highlight key components that collectively form the innovative approach to secure, transparent, and efficient data trading.

### 3.1. DataMesh+ Model Overview 

Figure 2 illustrates the operational overview of the proposed DataMesh+ model, which is a secure and reliable decentralized P2P data exchange model for SSDMs based on blockchain and decentralized storage technologies. This is a user-centric and decentralized approach that enables secure and reliable direct data trading transactions between peers without the need for a central authority or TTP. The DataMesh+ model leverages blockchain to ensure transaction integrity, transparency, traceability, and security. It uses smart contracts to automate fair trading processes and a decentralized P2P storage network to ensure that data remains secure and distributed, thus mitigating SPF concerns. In addition, it provides P2P communication channels (wallet-to-wallet, W2W) compatible with Ethereum addresses [59,60], enabling data sellers and buyers to negotiate and finalize their trades. In this context, the term *S* refers to a data owner who sells data (*Seller*), while *B* refers to a user who is willing to buy data (*Buyer*). The workflow of our proposed data trading approach depicted in Figure 2 unfolds as follows:①First, data sellers, denoted as *S* = {*S*_1_, *S*_2_, …, *S_n_*} create a set of data product profiles (*DPP*), defined as *DPP* = {*DPP*_1_, *DPP*_2_, …, *DPP_m_*}, which are indexed in the blockchain. These profiles are subsequently published on marketplaces to make them available for sale.②Any data buyer *B_i_* of a set *B* = {*B*_1_, *B*_2_, …, *B_k_*} can search or query a specific *DPP_id_* that matches her/his preferences.③The buyer sends a data product *DPP_id_* order request, ③’ which is automatically forwarded to the corresponding data seller *S_j_*.     Upon receiving the request, the seller confirms the order and sends an invoice. ④’ Simultaneously, a single-use decryption key and sample data are sent to the requester for verification and confirmation.④Upon receiving the sample data, the buyer verifies its correctness. The buyer can cancel the order request after deeming the data unsatisfactory.⑤If the buyer is satisfied with the sample data, he/she can make a payment, which is temporarily held in an escrow smart contract. The total amount paid (*Tot_a_*) is the sum of data price *P*, service commission fee (*C_φ_*), and transaction processing fee (*T_φ_*), as defined in Equation (9). *C_φ_* is the multiplication of data price *P* by *α* which is a predefined commission rate (i.e., *α* = 0.05%), as expressed by Equation (10). *T_φ_* is the sum of the transaction-related costs, as defined in Equation (11). ⑥’ After receiving the payment notification, the seller verifies the payment status.


*Tot_a_ = P + C_φ_ + T_φ_*
(9)


⑥Subsequently, the seller proceeds with the purchase and ships the purchased dataset to *B_i_*, the corresponding buyer.


*C_φ_ = α P*
(10)


⑦Upon receiving the purchased dataset, the buyer verifies its correctness and ⑧’ confirms data reception for the seller to receive the payment. The buyer can cancel the deal and request a refund if the dataset is unsatisfactory.⑧Upon confirming data reception, the smart contract releases funds and transfers *P* into the seller’s account.


(11)
$$T_{\varphi }=\sum_{i=1}^{n}\varepsilon_{\cos t}\left(T_{i}\right)$$


### 3.2. Architectural Framework

Figure 3 illustrates the multi-layered architectural framework of the proposed DataMesh+ model. The architecture is modular and segmented into five distinct layers, each serving specific functions within the marketplace platform.

The **Integration APIs and Service Layer** provides the necessary application programming interfaces (APIs), software development kits (SDK), and services that facilitate interoperability, accessibility, and interaction between the data trading system layer and external systems or services. This layer enables various applications and services from data import/export to interaction with the system, thereby facilitating a diverse range of data exchange services and improving platform usability.The **Secure and Reliable Data Trading System Layer** is a middleware that provides fair, transparent, secure, and trustworthy data trading features. It comprises the following principal modules: The *user profile manager* manages user profile credentials, roles, authentication, and authorization, ensuring accurate maintenance and secure access to user data. The *data product profile manager* allows the creation and management of data product profiles, which include metadata and usage terms associated with the datasets being traded. It also ensures accurate cataloging and retrievability of data products. The *data protection manager* enforces data protection policies, ensuring compliance with regulations and safeguarding data integrity. The *data search and query engines* facilitate efficient searching and querying within the marketplace, enabling users to find the data they need based on various search criteria. The *data discovery, import, and export* module facilitates the discovery, import, and export of data within the system, ensuring data integrity and accurate formatting during the process. The *data analytics manager* provides data processing and analytical capabilities, transforming raw data into actionable and valuable insights. The *data product review and recommender* collect user feedback on data products and recommend products to users based on their profiles, preferences, and past behavior. The *data quality and trust manager* maintains the data quality and manages trust scores for data products, ensuring adherence to high standards. The *data trading manager* facilitates data trading between users by overseeing transactions and enforcing the terms of agreed trade deals, including transaction validation, execution, and settlement. In addition, it provides features such as data product profile listing, price modeling, order management, and invoice management. The *payment gateway manager* handles financial transactions, allowing users to make payments for data products or receive payments within the marketplace. The *security and privacy manager* ensures that all transactions and data exchanges adhere to the highest security and privacy standards. The *decentralized access controller* ensures that data access is governed by decentralized policies, thereby enhancing security and user control.The **Blockchain Technology Layer** ensures the security, immutability, and transparency of transaction data. It facilitates the integration with the blockchain network for managing the distributed ledger and executing smart contracts. This layer comprises the following main components: The *smart contract manager* that govern the deployment, execution, and lifecycle of smart contracts governing data exchange agreements. The *transaction manager* acts as a facilitator within the blockchain, enabling the creation, validation, and processing of transactions. The *state DB* stores the current state of smart contracts and transaction data in a secure and accessible database. The *consensus manager* uses consensus algorithms to achieve agreement among network participants regarding the validity of transactions. The *blockchain manager* oversees the overall operation and maintenance of the blockchain network, including node management and network configuration. The *blockchain explorer* provides users with a graphical interface to explore and analyze blockchain data, including transaction history and smart contract details.The **Decentralized P2P Storage Layer** oversees a reliable IPFS-based decentralized data storage and access control across a P2P storage network. This layer comprises the following components: The *API gateway* serves as an interface for accessing and interacting with the IPFS network, allowing users to upload, retrieve, and manage data stored on IPFS. The *CID manager* manages content identifiers (CIDs) for data stored on IPFS, ensuring unique identification and retrieval of stored content. The *distributed hash table (DHT*) facilitates decentralized peer discovery and routing within the IPFS network, enabling efficient data retrieval and distribution. The *IPFS protocol* enables storage and sharing of data in a distributed and censorship-resistant manner, leveraging a network of peer nodes to ensure data availability and integrity.The **Secure Communication Infrastructure Layer** provides secure and reliable communication services built on dedicated secure Internet channels, such as scalability, control, and isolation on next-generation networks (SCION) [61]. SCION provides strong end-to-end encryption and protection against potential cyber threats [61]. Prioritizing scalability, control, and isolation, SCION enhances security compared with traditional Internet protocols. It uses path-aware routing and secure packet forwarding to mitigate common threats such as distributed denial-of-service (DDoS) attacks, route hijacking, and traffic analysis [61].

Together, these layers facilitate various functions within the system, leveraging blockchain technology for secure, transparent, and reliable market transactions. By using blockchain technology, the system ensures traceable and verifiable transactions, which promotes trust among users. In addition, P2P decentralized data storage solutions enhance the system’s resilience and mitigate the risks associated with centralized data storage. This architectural framework is designed to establish a secure environment for data trading that ensures reliability, compliance, and user sovereignty in data management.

### 3.3. Data Trading Algorithms

The data are encrypted before being stored off-chain in a secure, censorship-resistant, highly available P2P decentralized storage network such as IPFS, as depicted in Figure 4. First, the plaintext file uploaded by the user is encrypted to generate its ciphertext, which is then signed and tagged with metadata containing the file name, extension, owner address, hash value, and access control list identifier. The signed file with the public metadata is then stored on the IPFS network and the corresponding CID and access URL are recorded in a blockchain-based immutable and shared ledger. Data access is governed by a decentralized RBAC system driven by smart contracts and supported by the blockchain to ensure robust security features and efficient rights management to protect the privacy and integrity of data in multi-party environments. 

Listing 1 outlines the metadata schema of the data product profile. Algorithm 1 outlines the procedure for creating and publishing the proposed data product profile (*DPP*). It uses input parameters such as the seller’s identifier *S_j_* and *DPP* details, including the data offering profile *id*, name *η*, type *τ*, owner’s identifier *ω*, data hash *h*, digital signature *s*, previous owner’s identifier *υ*, access URL *ν*, price *p*, currency *c*, and status σ. The algorithm first confirms whether the sender of the message is the owner of the data and then checks whether *DPP_id_* does not exist in the blockchain to avoid duplicates. Subsequently, all input parameters are correctly assigned and transferred to the blockchain. After successful execution, the transaction hash and block number are returned.

**Listing 1.** Dataset profile data schema.struct Dataset {  string id;  string name;  string data_type;  string description;  string hash_value;  uint256 size;  string size_unit;  uint256 price;  string currency;  string cid;  string url;  address owner;  string signature;  address previous_owner;  bool isOpenForSell;  bool isOrdered;  bool isSold;   uint256 lastUpdate;}

**Algorithm 1:** Data product profile creation**Parameters**: Smart contract address *SC_a_*, account address *A_a_***Input**: Seller *S_j_* and *DPP* details as {*id*, *η*, *τ*, *ω*, *h*, *s*, *υ*, *ν*, *p*, *c*, *σ*} **Output**: *T_xh_*, *Block_no_*1:**if** *S_j_* = *ω*
**then**2:Check whether *DPP_id_* exists in the blockchain:3:*E* ← *sc.getDataset*(*DPP_id_*)4:**if** *E* ≠ NULL **then**5: Map each input parameter of *DPP_id_*:6: *DPP*[*_id_*] ← {*η,τ,ω, h, s,υ,ν, p, c,σ*}7: *DPP*[*_id_*].*isOpenForSell* ← *true*8: *DPP*[*_id_*].*isOrdered* ← *false*9: *DPP*[*_id_*].*isSold* ← *false*10:   *t* = *block.timestamp*11:   *DPP*[*_id_*].*lastUpdate* ← *t*12:   Push *DPP*[*_id_*] instance to the blockchain13:   Emit *NewDatasetAdded*(*DPP_id_*, *ω, σ, t*) event14:  **else**15:   Return “Dataset profile *DPP_id_* already exists.”16:  **end if**17: **else**18:  Return “Only owner/seller can register dataset.”19: **end if**20:  Return the transaction execution state (*T_xh_, Block_no_*).

Listing 2 shows the order data schema used by Algorithm 2, which represents the procedure for creating a data order. Algorithm 2 receives the order number *O_no_*, *DPP_id_*, and buyer identifier *B_i_* as inputs. It first checks whether the buyer’s identifier differs from the seller’s and ensures that *O_no_* does not already exist in the blockchain. Subsequently, all order parameters are saved, the states of the corresponding *DPP_id_* are updated, and the order data record is transferred to the blockchain. Algorithm 3 describes the payment procedure for an order that receives the order number *O_no_*, buyer identifier *B_i_*, and the payment amount *P_a_* as inputs. It first checks whether *B_i_* matches the buyer ID of the order *O_no_*, then determines whether the balance exceeds (or at least matches) the total payment amount and whether the specified *P_a_* matches this amount. Subsequently, it is determined whether the *DPP_id_* status is “*Ordered*”, “*Not Sold*”, “*Not Paid*”, and “*Not Cancelled*”, before transferring the paid amount to the address of the escrow smart contract.

**Listing 2.** Purchase order data schemastruct Order {  string order_no;  address buyer;  string dataset_id;  uint256 price;  string currency;  uint256 fee_cost;  uint256 total_amount;  address seller;   bool isPaid;  bool isConfirmed;  bool isCompleted;  bool isCancalled;  bool isRefundPaid;  uint256 lastUpdate;}

**Algorithm 2:** Data order creation**Parameters**: Smart contract address *SC_a_*, account address *A_a_***Input**: *O_no_*, *DPP_id_*, *B_i_*, *T_φ_***Output**: *T_xh_*, *Block_no_*1:**if** *B_i_* ≠ *DPP*[*_id_*].*ω* **then**2: Check whether *O_no_* exists in the blockchain:3: *E* ← *sc.getOrder*(*O_no_*)4: **if** *E* ≠ *NULL* **then**5:  Map each input parameter of *O*[*_no_*]:6:  *O*[*_no_*]. ← {*DPP_id_*, *B_i_*}7:  *O*[_no_].*owner* ← *DPP*[*_id_*].*ω*8:  *O*[*_no_*].*price ← DPP*[*_id_*].*P*9:  *O*[_no_].*currency ← DPP*[*_id_*]*.c*10:  *O*[_no_].*commissionFee ← DPP* [*_id_*]*.C_φ_*11:  *O*[_no_].*totalAmount ←* (*DPP*[*_id_*].*P + DPP* [*_id_*]*.C_φ_ + T_φ_*)12:  *O*[_no_].*paymentAddr ← SC_a_*13:  *O*[_no_].*isPaid ← false*14:  *O*[_no_].*isConfirmed ← false*15:  *O*[_no_].*isCompleted ← false*16:  *O*[_no_].*isRefunded ← false*17:  *t = block.timestamp*18:  Update *DPP_id_* status:19:  *DPP*[*_id_*].*isOpenForSell ← false*20:  *20. DPP*[*_id_*].*isOrdered ← true*21:  *DPP*[*_id_*].*lastUpdate ← t*22:  Push *O*[*_no_*] instance to the bsslockchain23:   Emit *NewOrder*(*O_no_*, *B_i_*, *DPP_id_*, *ω*, *O_s_, t*) event24:  **else**25:   Return “Dataset order *O_no_* already exists.”26:  **else**27:   Return “Only buyers can create orders.”28: Return the transaction execution state (*T_xh_*, *Block_no_*).

**Algorithm 3:** Order payment**Parameters:** Smart contract address *SC_a_*, account address *A_a_***Input:**
*O_no_, B_i_, P_a_* // *P_a_*: Paid amount **Output:**
*T_xh_, Block_no_*1: **if** *B_i_* = *O*[*_no_*]*.buyer*
**then**2:  **if** (*B_i_.balance* ≥ *O*[*_no_*]*.T ot_a_*) ∧ (*P_a_* = *O*[*_no_*]*.T ot_a_*) **then**3:   **if** (*DPP* [*_id_*]*.isOrdered* = *true*) ∧ (*DPP* [*_id_*]*.isSold* = *false*) **then**4:    **if** (*O*[*_no_*]*.isPaid* = *false*) ∧ (*O*[*_no_*]*.isCancelled* = *false*) **then**5:      Transfer *P_a_*to *A_a_*6:      *O*[*_no_*]*.isPaid* = *true*
7:      *t* = *block.timestamp*
8:      Update *DPP_id_* status:9:     *DPP* [*_id_*]*.isSold* ← *true*10:      *DPP* [*_id_*]*.lastUpdate* ← *t*11:     Emit *OrderPaid*(*O_no_, B_i_, t*) event12:    **else**13:   Return “*O*[*_no_*] status must be unpaid or not cancelled.”14:  **else**15:    Return “*O*[*_no_*] status must be ordered and not sold.”16:  **else**17:   Return “Insufficient balance or *P_a_*is not equal to *T ot_a_*”18:**else**19: Return “Different from order buyer account.”20:Return the transaction execution state (*T _xh_, Block_no_*).

After a successful payment transfer, Algorithm 3 updates the states of *DPP_id_* and *O_no_* on the blockchain. Algorithm 4 handles the procedure for confirming receipt of the order, with *O_no_* and *B_i_* as input parameters. It first checks whether *B_i_* is identical to the buyer’s identifier from the order *O_no_*. Subsequently, it checks whether the status of *O_no_* is “*Paid*”, “*Not Confirmed*”, “*Not Completed*”, and “*Not Cancelled*”, and then updates the status of *O_no_* as confirmed in the blockchain. Once the buyer confirms receipt of the order, the actual payment is transferred to the seller’s account, and the data ownership record is updated according to Algorithm 5, which uses the message sender’s address and the order identifier *O_no_* as inputs. The algorithm checks whether the specified address is an authorized account of the smart contract administrator. Next, it confirms whether the escrow payment account holds sufficient funds for the payment amount to be transferred *P* and whether the order status is “*Confirmed*”, “*Not Completed*” and “*Not Cancelled*”. If all conditions are satisfied, payment *P* is transferred to the seller’s account *S_j_*, followed by the transfer of *DPP_id_* ownership on the blockchain. Subsequently, the related order and *DPP* states are updated accordingly. Finally, the block number and transaction hash are returned as proof that the transaction was completed successfully.
**Algorithm 4:** Confirm order reception**Parameters:** Smart contract address *SCa*, account address *Aa***Input:**
*O_no_, B_i_***Output:**
*T _xh_, Block_no_*1:**if***B_i_* = *O*[*_no_*]*.buyer*
**then**2: **if** (*O*[*no*]*.isPaid* = *true*) ∧ (*O*[*no*]*.isConfirmed* = *false*) ∧  (*O*[*no*]*.isCompleted* = *false*) ∧ (*O*[*no*]*.isCancelled* = *false*) **then**3: *O*[*no*]*.isConfirmed ← true*4: Emit *OrderConfirmed*(*O_no_, B_i_, O_s_, t*) event5:**else**6:Return “Only *B_i_* can confirm this order.”7:Return the transaction execution state (*T _xh_, Block_no_*).

**Algorithm 5:** Payment transfer and dataset ownership update**Parameters:** Smart contract address *SC_a_*, account address *A_a_***Input:**
*O_no_*, *msg.sender***Output:**
*T _xh_*, *Block_no_*1:**if***msg.sender* = *A_a_* **then**2: **if**
*SC_a_.balance* ≥ *P*
**then**3:  **if** (*O*[*_no_*]*.isPaid* = *true*) ∧ (*O*[*_no_*]*.isConfirmed* = *true*) ∧   (*O*[*_no_*]*.isCompleted* = *false*) ∧ (*O*[*_no_*]*.isCancelled* = *false*) **then**4:    Transfer *O*[*_no_*].*P* to *O*[*_no_*]*.S_j_*5:    *prvOwner* ← *O*[*_no_*]*.DPP* [*_id_*]*.ω*6:    *t* = *block.timestamp*7:    Update *DPP_id_* status:8:    *DPP* [*_id_*]*.owner* ← *O*[*_no_*]*.B_i_*9:    *DPP* [*_id_*]*.previousOwner* ← *prvOwner*10:    *DPP* [*_id_*]*.isSold* ← *true*
11:    *DPP* [*_id_*]*.lastUpdate* ← *t*
12:     *O*[*_no_*]*.isCompleted* ← *true*13:    Emit *PaymentSent*(*O_no_*, *A_a_*, *O_s_*, *t*) event14:   **else**15:    Return “*O*[*_no_*] status must be paid and confirmed.”16:  **else**17:   Return “Insufficient balance.”18:**else**19:   Return “Only admin can perform this operation.”20:Return the transaction execution state (*T _xh_*, *Block_no_*).

In the case of dataset order cancellation, Algorithm 6 executes by receiving the order identifier *O_no_* and the buyer’s account address *B_i_* as inputs. It begins by validating whether the given account address matches the one included in the order, and then checks if the associated *DPP’*s status is currently “*Ordered*”, “*Not Completed*”, and “*Not Cancelled*” yet. Subsequently, the *DPP*’s status is updated as “*Cancelled*” on the blockchain. Upon successful order cancellation, the payment refund process is automatically initiated. Algorithm 7 highlights the process of refunding a payment to the buyer for a cancelled order. After receiving a call with parameters, which includes the message sender address and order identifier *O_no_*. Algorithm 7 verifies whether the message sender’s address is an authorized smart contract’s admin account. It then checks if the escrow payment account balance meets or exceeds the amount of payment *P* to be transferred and ensures that the order status is “*Paid*”, “*Not Completed*”, “*Not Cancelled*”, and no refund has been executed. If these conditions are met, payment *P* is refunded to the respective buyer account *B_i_*, and the related order and *DPP* states are updated on the blockchain accordingly. Finally, the transaction hash and block number are returned as a confirmation that the transaction was successfully executed. Figure 5 provides a summarized visualization of the proposed smart contract-based fair, secure, and reliable P2P data trading model for SSDMs.
**Algorithm 6:** Dataset order cancellation**Parameters:** Smart contract address *SC_a_*, account address *A_a_***Input:**
*O_no_*, *B_i_***Output:**
*T_xh_, Block_no_*1:**if** *B_i_* = *O*[*_no_*]*.buyer*
**then**2: **if** (*DPP* [*_id_*]*.isOrdered* = *true*) ∧ (*O*[*_no_*]*.isCompleted* = *false*) ∧   (*O*[*_no_*]*.isCancelled* = *false*) **then**3:  *O[_no_]*.isCancelled* ← *true**4:  Emit *OrderCancelled*(*O_no_*, *B_i_*, *O_s_*, *t*) event5: **else**6:   Return “*O*[*_no_*] status not be completed and cancelled.”7:**else**8:  Return: “Only *B_i_* can cancel the order.”9:Return the transaction execution state (*T _xh_*, *Block_no_*).

**Algorithm 7:** Payment refund to the buyer**Parameters:** Smart contract address *SC_a_*, account address *A_a_***Input:**
*O_no_*, *msg.sender***Output:**
*T_xh_*, *Block_no_*1:**if***msg.sender* = *A_a_* **then**2:**if***SC_a_.balance* ≥ *P*
**then**3: **if** (*O*[*_no_*]*.isPaid* = *true*) ∧ (*O*[*_no_*]*.isCompleted* = *false*) ∧  (*O*[*_no_*]*.isCancelled* = *true*) ∧ (*O*[*_no_*]*.isRefunded* = *false*) **then**4:  Transfer *O*[*_no_*].*P* to *O*[*_no_*]*.B_i_*5:  Update *DPP_id_* status: *DPP* [*_id_*]*.isOrdered* ← *false*6:  O[*_no_*]*.isRefunded* ← *true*7:  Emit *PaymentSent*(*O_no_*, *A_a_*, *O_s_*, *t*) event8:**else**9:Return: “Insufficient balance.”10:**else**11: Return: “Only admin can perform this operation.”12:Return the transaction execution state (*T_xh_*, *Block_no_*).

## 4. Implementation and Evaluation

This section encompasses the considerations for implementing the proposed model and the subsequent experimental evaluation.

### 4.1. Experimental Environment Setup and Performance Metrics

Table 2 shows the experimental environment setup and key performance metrics. The smart contracts of the proposed system were developed in the Solidity programming language. They were then deployed and tested in the Goerli test network. Goerli is a cross-client test network for the Ethereum blockchain [27] that uses the PoA consensus protocol. The PoA algorithm is well suited for permissioned blockchains, where known validators from different organizations oversee network management. It provides higher performance with *O*(*n*) computational complexity and can accommodate up to *f* faulty nodes within 2*f* + 1 consensus nodes (*n*) [54]. For payment, we use the native Ethereum blockchain cryptocurrency, *Ether* (*ETH*) [27], which is created through the mining process as a reward for peer nodes that secure the network. *MetaMask* wallet is used for transaction signing and verification. The DApp is built using various programming languages, libraries, and frameworks including React, node.js, hardhat, and web3.js APIs. Remote procedure calls (RPCs) facilitate interaction with Ethereum nodes through smart contracts. Infura APIs provide secure, reliable, and scalable access to Ethereum and IPFS networks. To prevent network abuse issues and reward resource-providing nodes, Ethereum imposes fees for every programmable computation [27]. These fees, which are denominated in units of gas with equivalence in Ether, and cover various computations such as contract creation, account storage, state updates, and other execution of EVM operations [27,52]. 

### 4.2. Operational Cost Evaluation

To assess the feasibility and reliability of our model, we evaluated the deployment and operational costs of core smart contracts by considering the following parameters:*Smart contract deployment cost*: The deployment cost encompasses the code deposit, execution, and transaction costs. The code deposit cost is the maximum amount of gas required for successful contract creation by placing the code into the state [27,52]. It varies with the size of the bytecode generated from the compiled contract source code. Table 3 presents the measurements of core smart contract deployment gas costs, including dataset management and data trading management contracts. Deployment cost is expressed in Gwei, Ether, and USD, with a gas price of 1 ETH = $1.283.23 (2022.12.09). Although gas prices fluctuate, we used this price for simplicity in our evaluation. Our smart contracts were optimized for cost-effectiveness, with the deployment cost for *DatasetMgr.sol* and *DataTradingMgr.sol* smart contracts being approximately 0.00234 Ether ($3.03) and 0.0052 Ether ($6.73), respectively.

*Execution cost*: The execution cost is the overall computational gas cost to execute transaction operations using an EVM, as defined in Equation (12).


(12)
$$\varepsilon_{\cos t}=\sum_{i=1}^{n}{\mathrm{OP}}_{i}$$


*Transaction cost*: Transaction costs, also referred to as gas fees or the amount of gas used (*G_cost_*), are fees paid by users to miners for processing transactions and ensuring the security of the the blockchain network [27,52]. These fees play a crucial role in resource allocation and network stability. The transaction gas fee (*T_fee_*) is calculated by multiplying the amount of gas used *G_u_* by the gas price *G_p_*, as shown in Equation (13). It is worth noting that transaction costs on the blockchain can fluctuate because of factors such as network congestion, gas prices, and the complexity of the smart contract functions involved in the transaction [52]. Table 4 provides a summary of the operational transaction execution gas costs in Gwei, Ether, and USD. The experimental results indicate that, on average, the transaction execution cost is 0.0001 Ether (0.14 USD) for writing operations and 0.000029 Ether (0.04 USD) for reading operations. Tasks that primarily involve retrieving data (getter functions) generally incur lower fees because they do not change the state of the ledger. Conversely, tasks that involve frequent data updates (setter functions) are likely to incur higher costs depending on their complexity.


(13)
$$T_{\mathrm{fee}}=G_{u}*G_{p}$$


### 4.3. Blockchain and IPFS Performance

Using the PoA consensus protocol, new blocks are added to the Ethereum blockchain every 12s. A block is identified by an index that corresponds to the block number and height of the blockchain. It also contains several parameters, including: the *timestamp,* which indicates when the block was proposed; the *transaction number,* which reflects the number of transactions included in the block; the *size*, which denotes the data size in bytes; the *gas limit,* which represents the maximum gas set by the transactions in the block; and the *gas used,* which is the total units of gas used by transactions in the block. Figure 6a provides measurements of block size per index for 120 sampled blocks, with an average block size of approximately 102.34 KB. The number of transactions per block varies depending on several factors, including the block size limit, the rate at which blocks are produced, and the number of transactions the network processes per unit of time. In Figure 6b, we assessed the number of transactions included in every block, which is 60 on average.

Figure 7a shows the gas usage for transactions and blocks processed by the network, with the average block creation and confirmation depending on the gas price. Notably, contracts that consume gas contribute significantly to network usage, with an average of 15,736,700 gas units used per block, which corresponds to the total of gas used by transactions included in the block. Figure 7b shows the relationship between block timestamps, epochs, and slot time variations. The *epoch* parameter indicates the epoch in which the block was proposed, and the *slot* corresponds to the slot in which the block was proposed. Slots refer to opportunities for block creation (adding one valid block). The block timestamp and slot scaled linearly as the block index increased, whereas the block epoch scaled gradually. This resulted because several slots existed in this epoch. The average block time spans approximately 15 s.

Transaction throughput is the number of transactions processed per unit of time, typically measured in transactions per second (TPS). In contrast, transaction latency represents the time taken for a transaction to be processed and included in a block. These metrics in a blockchain network depend on various factors such as network usage, block size, time, and network congestion. As shown in Table 2, the blockchain network exhibited an average transaction throughput and latency of 24 TPS and 36 s respectively, considering the need for at least three block confirmations. Furthermore, the average encryption and storage times per *DPP* object of 38 KB size were 25.4 and 38.6 ms, respectively. The experimental evaluations were performed on a server with an 11th Generation Intel(R) Core*^TM^*i7-11700 processor (2.50GHz), 64 GB of RAM, and Gigabit Ethernet network interfaces.
(14)$$B_{u}=\frac{\mathrm{ADT}}{\mathrm{MTR}*\mathrm{TP}}$$

The IPFS network bandwidth use (*B_u_*) is the ratio of the amount of data being transferred per unit of time to the maximum data transfer rate, as computed using Equation (14). *ADT* denotes the amount of data being transferred that corresponds to the sum of incoming (*B_in_*, in bandwidth) and outgoing (*B_out_*, out bandwidth) data transfer rates, as expressed in Equation (15). *MTR* denotes the maximum transfer rate achievable for the network, and *TP* is the duration of data transfer.
(15)$$\mathrm{ADT}=B_{\mathrm{in}}+B_{\mathrm{out}}$$

Figure 8 presents the global IPFS network traffic bandwidth use measurements over time demonstrating the network’s strong stability and excellent performance with minimal bandwidth consumption. Specifically, the total data transferred comprises 1.2 GB (incoming) and 679 MB (outgoing), with average data rates of 109.70 KB/s (in) and 83.12 KB/s (out).

### 4.4. Security Analysis

This section assesses the security properties of the proposed model, encompassing the following:*Entity and data origin authentication*: Participant entities are identified by Ethereum EOAs, which are authenticated using public-private key pairs and digital signatures [27,52]. Each party possesses a unique private key that is used to generate digital signatures. Verification of the digital signature using the corresponding public key allows the receiving party to authenticate the sender’s identity. The data origin is authenticated and verified through ownership and signature proofs stored on the blockchain.*Data confidentiality and security*: Blockchain and IPFS technologies ensure data integrity and non-repudiation security [27,29,52]. Digital signatures uphold the integrity of the traded data; any alteration to the data would invalidate the signature [52,53]. In addition, they ensure non-repudiation, preventing the sender from denying sending the data. Once signed and transmitted, the signature serves as an irrefutable proof of consent. However, as a public blockchain, neither Ethereum nor IPFS provides data confidentiality. To address this issue, the data are encrypted before storage in IPFS, while the metadata are stored in the blockchain. The proposed system mitigates replay attacks in which the same signature is repeatedly used (“replayed”) for unauthorized actions [52,62]. For example, a payee resubmitting a signature to claim a second payment (double spending attack) poses a serious security risk [62,63]. The security model of the PoA consensus algorithm integrates digital signatures and a tamper-proof ledger to ensure that historical records remain untampered with [54]. This prevents malicious actors from forging payments or falsely reporting asset transfers.*Accountability, transparency, and fairness*: Each participant is held accountable for their actions through activity history recorded in a cryptographically secure and tamper-proof blockchain-based ledger [22,26,27]. This ledger serves as an immutable record that fosters transparency and traceability for every transaction. It ensures that participants’ actions are transparent and traceable, thus fostering a culture of accountability [40,47,48,49,50,51]. To prevent data misuse and maintain fairness, trade history—encompassing ownership transfers and usage terms—is securely stored on the blockchain. This approach facilitates continuous monitoring and auditing of activities, promoting fairness in the data exchange process. Furthermore, the blockchain-based decentralized P2P storage network enhances transparency, prevents fraud, fosters trust among participants, and promotes ethical behavior within the data marketplace ecosystem.*High availability and reliability*: The integration of blockchain and IPFS networks with our proposed system guarantees censorship resistance and robustness, ensuring globally consistent availability and high reliability [29,54,55,56,57,58]. However, the gas fee required to perform transactions introduces a risk of insufficient gas [62,63,64,65], which may affect users’ ability to operate within the system. To mitigate this issue, users are advised to maintain a sufficient balance in their system operating accounts.*Smart contract vulnerabilities*: Ensuring that smart contracts do not contain bugs or security flaws is crucial before their deployment on the blockchain because they cannot be patched or modified once they are deployed [52,63,66]. Our developed *DatasetMgr.sol*^3^ and *DataTradingMgr.sol*^4^ smart contracts were successfully analyzed using SOOHOOdin^5^, a state-of-the-art smart contract security vulnerability analyzer. As shown in Figure 9, they are robust against up-to-date smart contract vulnerabilities, including reentrancy, integer overflow/underflow, unchecked external calls, unprotected ether withdrawal, and gas limit issues [62,63,64,65,66].

Next, we present a theoretical analysis of the attack scenarios and mitigation strategies using the proposed framework.

*Reentrancy attacks*: Malicious actors may exploit reentrancy vulnerabilities in smart contracts to repeatedly invoke functions, potentially withdrawing funds or causing unexpected behavior [62,63]. With the proposed model, such reentrancy attacks can be mitigated by implementing secure coding practices, e.g., using the "*Checks-effects-interactions*" pattern [63,64,65,66] to ensure that state changes are made before interacting with external contracts or transferring funds.*Front-running attacks*: Malicious actors can execute front-running attacks by monitoring pending transactions and strategically submitting their transactions to exploit price changes or manipulate the transaction order [52,67]. To mitigate such attacks with the proposed model, mechanisms such as commit-reveal schemes or encrypted order submissions can be implemented to obscure transaction details until execution on the blockchain, preventing attackers from preempting legitimate transactions.*Smart contract bugs*: Bugs or logical vulnerabilities in smart contracts can be exploited by attackers to bypass access controls, manipulate data, or cause unexpected behavior in the system [64,65,66]. These risks can be mitigated through comprehensive code reviews, the use of formal verification techniques, and the introduction of bug bounty programs that incentivize security researchers to identify and report vulnerabilities.*Sybil Attacks*: In a Sybil attack, malicious entities create multiple fake identities to gain control over a significant portion of network resources and influence system behavior [68]. To mitigate Sybil attacks, robust identity verification mechanisms, such as proof-of-individuality protocols or reputation systems, can be implemented to prevent the propagation of fake identities and maintain network integrity.

## 5. Discussion

In this section, we discuss the potential limitations of the proposed approach and the remaining open challenges, including the following:*Transaction gas fees*: Transaction fees, particularly in blockchain networks such as Ethereum, pose a major challenge because of their unpredictability and volatility [27,52]. During periods of heightened network activity, gas fees surge as miners compete for limited block space. Consequently, transaction costs escalate, affecting user experience and increasing the risk of transactions running out of gas mid-execution. Such occurrences can result in transaction failures or partial execution, leading to resource wastage and frustrating user experiences [62,63,64,65]. Effective mitigation strategies involve meticulous estimation of gas limits and costs before transaction execution and optimization of smart contract code to ensure efficient gas usage [63,66].*Smart contract legal design and upgradability considerations*: Smart contract design should include dispute resolution considerations covered by legal regulations. In contrast to traditional software, smart contracts lack upgradability once they are deployed on the blockchain [52,63,64,65]. This limitation poses challenges in adapting to evolving legal frameworks, particularly in e-commerce and related marketplaces. Although smart contracts enhance efficiency and transparency, they also present the challenge of adapting to unforeseen circumstances [32,64]. Predefined terms can restrict their flexibility, leading to suboptimal outcomes in dynamic trading environments. Developing smart contract-driven reliable and fair protocols requires meticulous coding and a deep understanding of the underlying business logic to minimize loopholes or unintended consequences [63,64]. External factors such as market manipulation or regulatory changes exacerbate smart contract limitations. Addressing these concerns requires innovative smart contract design to improve adaptability and resilience to external influences. Thus, exploring complementary mechanisms such as DApps, rigorous testing, and auditing processes to mitigate potential risks remains an area of future research interest.*P2P data marketplace governance*: The absence of a central authority in the proposed P2P data marketplace system necessitates efficient and reliable decentralized governance and trading moderation protocols. Challenges arise from the lack of global data protection regulations, varying jurisdictions, and governance structures governing data sovereignty in P2P data marketplaces.*Data quality assessment methods*: Optimizing data quality is essential for pricing and trade reliability [69,70]. Implementing reliable and efficient data quality assessment techniques is imperative for optimizing the quality of the traded data in the proposed system model.*Pricing model*: Although this study employs a negotiated price approach between data sellers and buyers, dynamic pricing models such as auction-based [22,49], data quality-based [69,70], or market-driven [71,72] approaches are viable alternatives.*Decentralized review, trust, and reputation management schemes*: Efficient decentralized review, trust, and reputation management schemes are essential to leverage the proposed system operating in a trustless environment [73,74,75,76,77,78].*Efficient incentive distribution mechanism*: Given that the proposed P2P data marketplace system model relies on community members, implementing novel, secure, and efficient incentive distribution mechanisms is vital [77,78].*Computing resource and energy consumption*: The PoA consensus algorithm is renowned for its efficient computing resource and energy consumption scheme [54], which drives its widespread adoption. In contrast, the PoW consensus algorithm is slower and requires considerable computing resources and energy to solve complex mathematical problems [26,27]. In addition, IPFS network nodes exhibit low computation overhead and bandwidth consumption.*Network scalability*: The system’s scalability depends on the underlying blockchain and IPFS networks, both grappling with scalability challenges [27,28,29,55,56]. Balancing scalability, decentralization, and security is essential. Addressing these challenges involves improvements in blockchain scalability and smart contract code optimization, as well as the development of layer 2 scaling solutions and protocol upgrades. These measures mitigate network congestion, reduce gas fees, and enhance network efficiency and throughput.

## 6. Conclusions and Future Work

This paper proposes DataMesh+, an innovative, secure, and reliable P2P data exchange model for decentralized self-sovereign data marketplaces, which is underpinned by a robust infrastructure powered by blockchain and decentralized storage technologies. In this model, smart contracts autonomously execute transactions based on pre-agreed terms between buyers and sellers. DataMesh+ represents not only an advancement of the data mesh architectural framework but also a paradigm shift in the operational dynamics of data exchange within marketplaces. This approach differs from traditional data marketplaces in that it prioritizes user-oriented, fair, transparent, secure, and reliable transactions. It enables trustworthy and auditable exchanges between anonymous parties worldwide without relying on centralized trusted third parties. Users retain control over their traded data assets using strong cryptographic techniques, including public-private key pairs and digital signatures, to ensure authentication and accountability. All transactions are recorded in a blockchain ledger, which is known for its tamper-proof and immutable properties, thereby guaranteeing transparency and traceability. Data confidentiality is also prioritized and ensured by robust encryption methods. Furthermore, this paper provides a comprehensive review of the literature on decentralized, blockchain-based P2P data marketplaces. This study explores background research, design considerations, operating principles, and challenges in implementing such systems. The feasibility of the proposed model is demonstrated through a prototype supported by experimental testing and validation, which confirms its reliability and effectiveness. Looking forward, several areas warrant future research. These include the development of reliable methods for assessing data quality, dynamic pricing models, decentralized systems for managing trust and reputation, and efficient governance and incentive distribution protocols for P2P data marketplaces. In addition, empirical studies or surveys to further assess user perception of how the proposed approach enhances user experience and data sovereignty should be conducted. These areas represent crucial steps in the evolution of decentralized data trading platforms, potentially enhancing their efficacy and application.

## 7. Patents

In, H.P.; Merlec, M. M. Blockchain-based safe and reliable data transaction method, and data transaction platform providing system. *WIPO (PCT)*, WO2022145679A1, 7 July 2022.In, H.P.; Merlec, M. M. Blockchain-based secure and trusted data trading methods and platform system. *Korean Patent*, KR102540415B1, 5 June 2023.

## Figures and Tables

**Figure 1 sensors-24-01896-f001:**
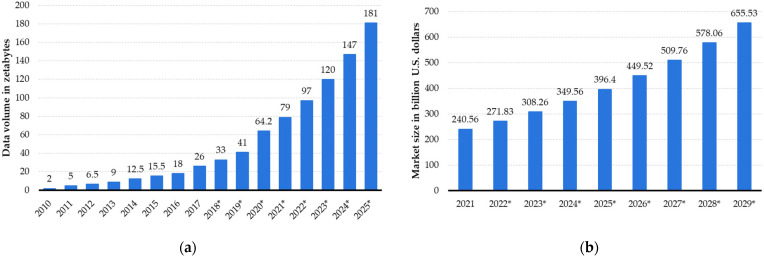
(**a**) Amount of data created, collected, and used worldwide from 2010 with predictions up to 2025 (in zettabytes) [1]; (**b**) Global big data analytics market size from 2021 to 2029 (in billions of US dollars) [2].

**Figure 2 sensors-24-01896-f002:**
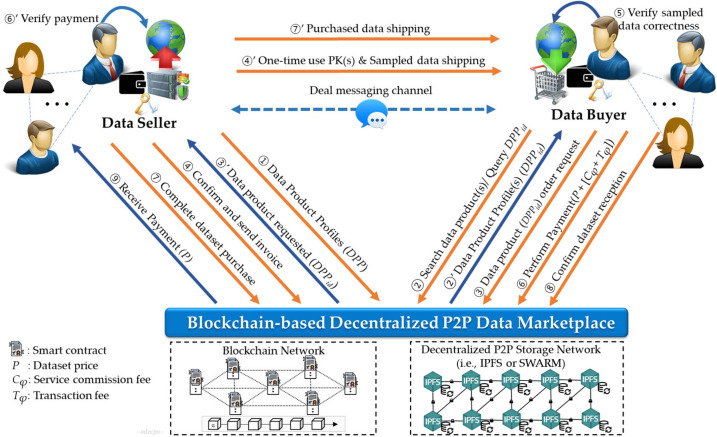
Overview of the proposed blockchain-powered P2P data exchange model (*DataMesh*+) for decentralized self-sovereign data marketplaces (SSDM).

**Figure 3 sensors-24-01896-f003:**
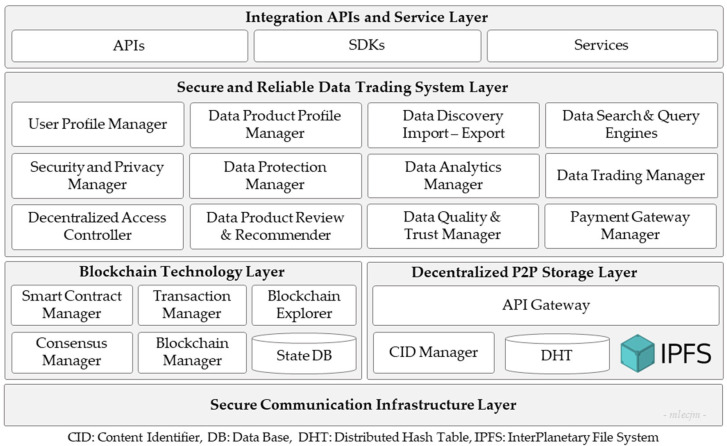
The architectural framework of the proposed blockchain-enabled P2P data exchange model for decentralized self-sovereign data marketplaces.

**Figure 4 sensors-24-01896-f004:**
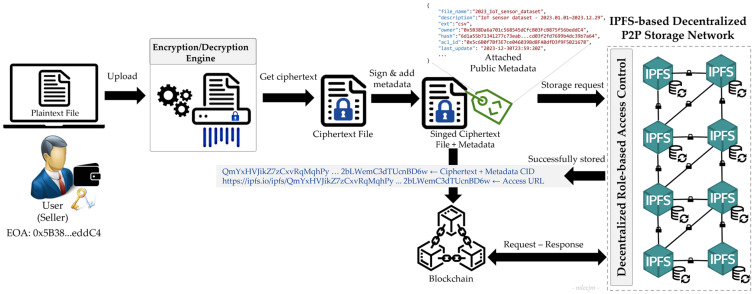
Data encryption before being stored in an IPFS-based decentralized P2P storage network.

**Figure 5 sensors-24-01896-f005:**
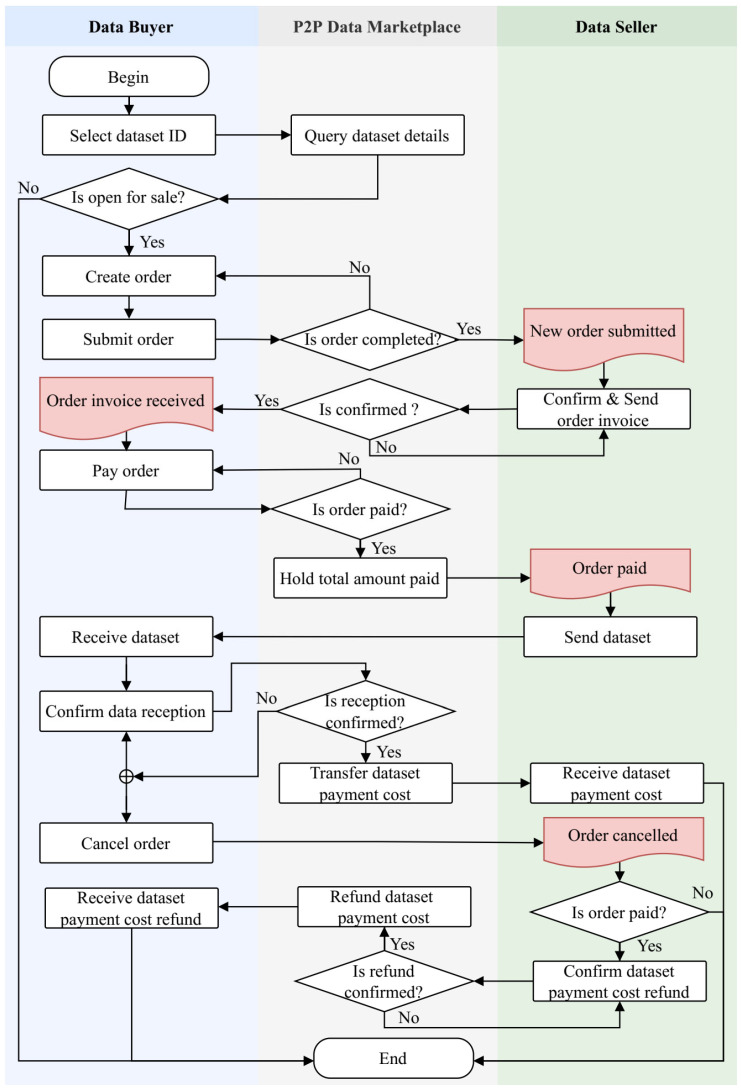
The smart contract-based fair, secure, and reliable P2P data trading flowchart of the proposed model for SSDMs.

**Figure 6 sensors-24-01896-f006:**
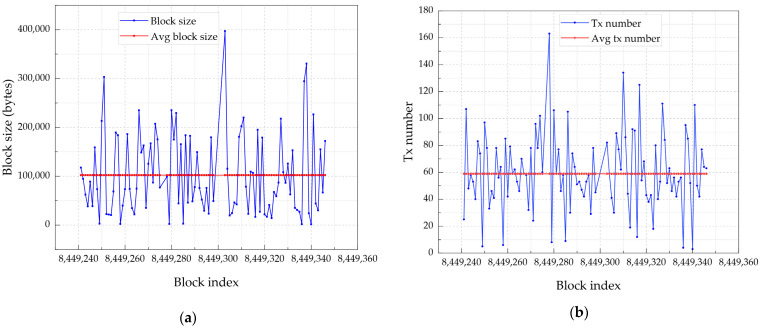
Block size and number of transactions per block index: (**a**) Block size measurements per index for 120 blocks; (**b**) Number of transaction measurements per index for 120 blocks.

**Figure 7 sensors-24-01896-f007:**
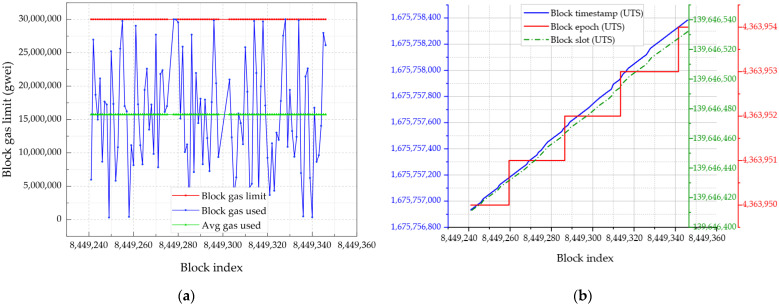
Gas spent, block timestamp, slot, and epoch time per block index: (**a**) Gas limit and gas spent measurements per index for 120 blocks; (**b**) Block timestamp, slot, and epoch time measurements per index for 120 blocks.

**Figure 8 sensors-24-01896-f008:**
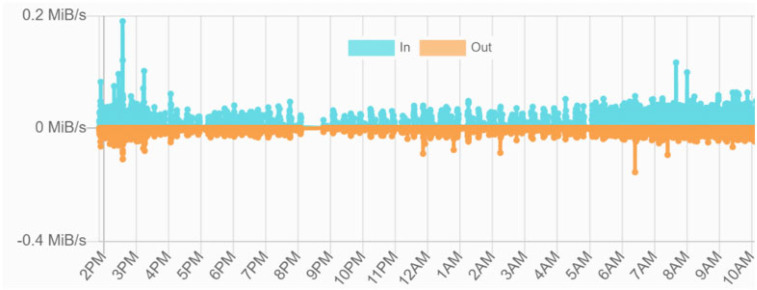
IPFS network traffic in-out-bandwidth utilization measurements.

**Figure 9 sensors-24-01896-f009:**
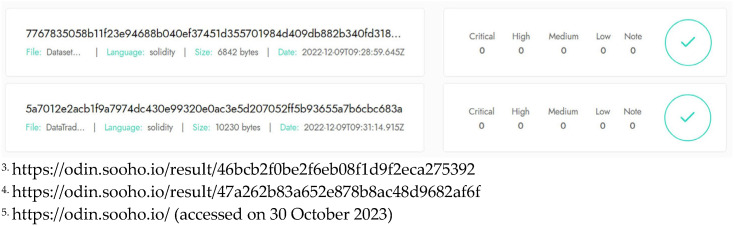
Smart contract security vulnerability analysis reports.

**Table 1 sensors-24-01896-t001:** Comparison of the proposed approach with previous works.

Ref./ Features	Data Mesh	P2P ^1^	Blockchain	DDM ^2^	SC ^3^	IPFS ^4^	W2WM ^5^	COC ^6^	DACS ^7^	S & T ^8^	T & T ^9^	Fairness	Accountability	Built PE ^10^	
[4]	✗	✗	✗	✗	✗	✗	✗	✗	✗	✗	✗	✗	✗	✓	✓
[6]	✗	✗	✗	✗	✗	✗	✗	✗	✗	✗	✗	✗	✗	✓	✓
[7]	✗	✗	✗	✗	✗	✗	✗	✓	✗	✓	✗	✗	✗	✓	✓
[8]	✗	✗	✗	✗	✗	✗	✗	✗	✗	✗	✗	✗	✗	✓	✓
[12]	✗	✗	✓	✗	✓	✗	✗	✗	✗	✓	✓	✗	✓	✓	✗
[13]	✗	✓	✓	✗	✓	✗	✗	✗	✗	✓	✓	✗	✓	✓	✓
[14]	✗	✓	✓	✓	✓	✗	✗	✓	✗	✓	✓	✗	✓	✓	✗
[15]	✗	✓	✓	✓	✓	✓	✗	✗	✗	✓	✓	✗	✗	✓	✗
[34]	✗	✓	✓	✓	✓	✗	✗	✗	✗	✗	✓	✗	✓	✓	✓
[35]	✗	✓	✓	✗	✓	✗	✗	✗	✓	✓	✗	✗	✓	✗	✗
[37]	✗	✗	✓	✓	✓	✗	✗	✗	✓	✓	✓	✗	✓	✗	✗
[39]	✗	✓	✓	✓	✗	✗	✗	✓	✗	✓	✓	✗	✗	✓	✓
[40]	✗	✓	✓	✗	✓	✗	✗	✗	✗	✓	✓	✗	✓	✓	✓
[41]	✗	✗	✓	✓	✓	✓	✗	✗	✗	✓	✓	✗	✓	✓	✓
[42]	✗	✗	✓	✓	✓	✗	✗	✗	✗	✓	✓	✓	✓	✓	✓
[43]	✗	✗	✓	✓	✓	✗	✗	✗	✗	✓	✓	✗	✓	✓	✗
[45]	✗	✗	✓	✗	✓	✗	✗	✗	✗	✓	✓	✗	✗	✓	✗
[46]	✗	✗	✓	✓	✓	✗	✗	✗	✗	✓	✓	✗	✓	✗	✗
[47]	✗	✗	✓	✓	✓	✗	✗	✓	✓	✓	✗	✗	✓	✓	✗
[48]	✗	✗	✓	✓	✓	✗	✗	✗	✗	✓	✓	✗	✓	✓	✓
[49]	✗	✓	✓	✓	✓	✗	✗	✗	✗	✓	✓	✗	✓	✓	✓
[51]	✗	✓	✓	✓	✓	✗	✗	✓	✗	✓	✓	✗	✓	✓	✗
This work	✓	✓	✓	✓	✓	✓	✓	✓	✓	✓	✓	✓	✓	✓	✓

^1^ Peer-to-Peer, ^2^ Decentralized Data Marketplace, ^3^ Smart contract, ^4^ InterPlanetary File System, ^5^ Decentralized Access Control System, ^6^ Consent & Ownership Control ^7^ Wallet-to-Wallet massaging system for deal negotiation, ^8^ Security & Trust, ^9^ Transparency and Traceability, ^10^ Performance evaluation.

**Table 2 sensors-24-01896-t002:** Experiment environment setup and performance metrics.

Parameters	Values
Network Network ID Chain ID Consensus Protocol Number of nodes Epoch interval Step period Total difficulty Gas limit Average transaction size (bytes) Transaction throughput (TPS) Average transaction latency (sec) Average block size (KB) Average number of Tx per block Number of block confirmations Average block time (sec) Blockchain network utilization (%) Smart contract language Compiler version EVM Version Digital wallet Dapp frameworks and IPAs Average encryption time per DPP (ms) IPFS node & kubo agent IPFS storage time per DPP object (ms) Average size per DPP object (kb) IPFS network bandwidth usage (kb/s)	Goerli Testnet (https://goerli.net/ accessed on 9 December 2022) 5 5 PoA Clique 30 30,000 15 10,790,000 30,000,000 3760 24 36 102.34 60 16 15 61.40 Solidity v0.8.17 London MetaMask v10.23.3 React, nodejs, hardhat, web3.js 25.4 v0. 26.0 & v0.18.1 39.6 38 109.70 (in)/83.12 (out)

**Table 3 sensors-24-01896-t003:** Experimental environment setup and performance metrics.

No	Smart Contract	Deployment Cost		
		Gas Used (Gwei)	ETH	USD ^†^
(1) (2)	DatasetMgr.sol ^1^ DataTradingMgr.sol ^2^	2,361,016 5,243,074	0.00236102 0.00524307	3.029726562 6.728069849

^1^ Deployed contract address: 0x0d77e6a61c7fb5af4c0c332a524e8b70619f0e49 ^2^ Deployed contract address: 0x1d682c7098cd34748990925b1bc1651c9cff91eb ^†^ ETH Price: 1 ETH = $ 1283.23 (9 December 2022)—https://coinmarketcap.com/.

**Table 4 sensors-24-01896-t004:** Operational gas costs of core smart contract-related functions.

No	Smart Contract	Operational Cost		
		Gas Used (Gwei)	ETH	USD ^†^
(1)	newDataSet	406,619	0.00040662	0.52178570
(2)	getDatasetDetail	53,166	0.00005317	0.06822421
(3)	isOpenForSale	25,499	0.00002550	0.03272108
(4)	openForSale	48,897	0.00004890	0.06274610
(5)	isSold	25,478	0.00002548	0.06274610
(6)	closeSale	26,953	0.00002695	0.03458690
(7)	getDataPrice	29,117	0.00002912	0.03736381
(8)	changePrice	44,198	0.00004420	0.05671620
(9)	getOwner	25,493	0.00002549	0.03271338
(10)	changeOwnership	44,155	0.00004416	0.05666102
(11)	newOrder	287,227	0.00028723	0.36857830
(12)	getOrderDetail	41,097	0.00004110	0.05273690
(13)	getOrderAmount	29,096	0.00002910	0.03733686
(14)	isPaid	25,454	0.00002545	0.03266334
(15)	payOrder	48,800	0.00004880	0.06262162
(16)	confirmReception	36,633	0.00003663	0.04700856
(17)	sendPayment	74,048	0.00007405	0.09502062
(18)	isRefunded	25,500	0.00002550	0.03272237
(19)	refundOrder	76,039	0.00002550	0.03272237
(20)	withdraw	35,185	0.00007405	0.09502062
(21)	getBalanceOf	22,234	0.00002223	0.02853134

^†^ ETH Price: 1 ETH = $ 1283.23 (9 December 2022)—https://coinmarketcap.com/.

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
