# Peer review of "DataMesh+: A Blockchain-Powered Peer-to-Peer Data Exchange Model for Self-Sovereign Data Marketplaces"

_sensors, 2024, doi:10.3390/s24061896_

Round 1
Reviewer 1 Report
Comments and Suggestions for Authors
There are several suggestions:
(1)The authors could give more detailed examination of potential vulnerabilities, including smart contract security flaws (e.g., reentrancy, gas limit issues) and explain how the model mitigates such risks. If authors can show some case studies or theoretical analysis of attack scenarios would also be valuable.
(2)The authors also emphasizes a user-centric approach but could further detail how it enhances user experience and data sovereignty. Including user studies or surveys on usability, user control features, and data ownership perceptions could provide empirical evidence of the model's user-centric benefits.
(3)Please check carefully and correct all the typos and equation format in the revision.
Comments on the Quality of English LanguageEnglish could be further improved, and all the grammar and composition mistakes should be corrected in the revision.
Author Response
First of all, we would like to thank the reviewer for appreciating our work and providing helpful comments to improve the quality of the paper. Please find in the attached file the cover letter of our point-by-point responses to the reviewers’ comments. We hope that the reviewer will find our modifications (in blue color font) and explanations satisfactory.

Reviewer 2 Report
Comments and Suggestions for Authors
· It is recommended to simplify the manuscript title.
· Reduce the number of keywords.
· I don’t agree with the statement “Digital signatures authenticate the identity of participants and hold them accountable.” Digital signatures can contribute to accountability, but they don't directly hold participants accountable. While they provide a way to link an action to a specific identity, enforcing consequences for that action requires additional mechanisms.
· Section 2 should be simplified.
· The authors used smart contracts to automate fair trading processes. Discuss the limitations of smart contracts such as:
o Fair trading requires anticipating unforeseen circumstances. Smart contracts, however, rely on pre-defined conditions and may not adapt well to unexpected situations.
o Creating truly fair and comprehensive smart contracts requires complex and meticulous coding, leaving room for potential loopholes or unintended consequences.
o Fair trading can be influenced by external factors beyond the scope of a smart contract, such as market manipulation or regulatory changes.
· Provide enough technical details on public-private key pairs, the used digital signature method, the used hash method, and encrypting the data before being saved in IPFS.
· For Section 3.2, provide technical details enough to understand the proposed method. The authors provided general information, without enough technical information.
· Provide examples of metadata associated with the datasets being traded.
· Improve the manuscript's general format.
· Explain the SCION secure internet architecture.
· Write the full term and then its abbreviation, such as RPCs (remote procedure calls).
· For page 17, elaborate on the following points:
o Tasks primarily involving data retrieval will generally incur lower fees.
o Tasks involving frequent data updates will likely have higher associated costs.
o Actual transaction costs can fluctuate based on network congestion and other factors.
o Future blockchain innovations may potentially reduce transaction costs.
Comments on the Quality of English Language· The manuscript requires major English proofreading.
· Turnitin indicates a 23% similarity index.
Author Response

(The authors gave the same response as above.)
